# The Correlation of LVI Parameters and CAI Behaviour in Aluminium-Based FML

**DOI:** 10.3390/ma16083224

**Published:** 2023-04-19

**Authors:** Piotr Podolak, Magda Droździel-Jurkiewicz, Patryk Jakubczak, Jarosław Bieniaś

**Affiliations:** Department of Materials Engineering, Faculty of Mechanical Engineering, Lublin University of Technology Nadbystrzycka 36, 20-618 Lublin, Poland; m.drozdziel@pollub.pl (M.D.-J.); p.jakubczak@pollub.pl (P.J.); j.bienias@pollub.pl (J.B.)

**Keywords:** fibre metal laminate, compression after impact, damage analysis, digital image correlation, residual strength, damage tolerance

## Abstract

An experimental analysis of mechanical behaviour for aluminium-based fibre metal laminates under compression after impact was conducted. Damage initiation and propagation were evaluated for critical state and force thresholds. Parametrization of laminates was done to compare their damage tolerance. Relatively low-energy impact had a marginal effect on fibre metal laminates compressive strength. Aluminium–glass laminate was more damage-resistant than one reinforced with carbon fibres (6% vs. 17% of compressive strength loss); however, aluminium–carbon laminate presented greater energy dissipation ability (around 30%). Significant damage propagation before the critical load was found (up to 100 times the initial damaged area). Damage propagation for assumed load thresholds was minor in comparison to the initial damage size. Metal plastic strain and delaminations are dominant failure modes for compression after impact.

## 1. Introduction

As the demand for light and durable materials in aerospace increases, the growth of interest in composite materials is also visible [1]. Fibre-reinforced polymers (FRPs) are claimed to be the most common material in the construction of modern planes [2] (e.g., Boeing 787 [3,4]). This is supported by their excellent strength, low density and fatigue resistance [5]. Due to their laminar structure, FRPs are susceptible to crack growth between adjacent layers, which develop finally into delaminations. Low-velocity impact (LVI) events, which occur often in conditions of aircraft exploitation [6] are one of the sources of crack initiation and propagation mechanisms [7].

Among others, to improve the impact resistance of light aerospace structures, fibre metal laminates (FMLs) were introduced in the late 1980s [8]. The supplementation of FRP properties with the presence of metal with different mechanical behaviour [9] allows the limitation of impact-induced damages significantly and raise the amount of absorbed energy [10], which is proven by many studies [11]. To increase the impact resistance and energy absorption ability further, fibre metal laminates are used in some cases as face sheets for lightweight sandwich structures [12,13]. Among the most extensively studied and developed FMLs, are aluminium–glass fibre metal laminate (GLARE), used so far on an industrial scale [14], and carbon fibre-reinforced aluminium laminate (CARALL) as counterpart with greater strength and lower density [15,16].

The damage done by an LVI event is an essential factor for residual mechanical properties of composite materials [11]. The delaminations, matrix failure or fibre failure in the FRPs and FMLs may negatively change their response. Due to the action of load assumed for the intact structure, the existing flaws may propagate into more complex and far greater forms, leading to the premature critical failure of such components [17]. Such behaviour was particularly noted for axial compression: the LVI-induced damage led to up to 60% [18,19,20] loss of initial strength of FRP based on the laminae type or fibre arrangement used [21]. After subjecting FRPs to compression after impact (CAI), the delaminations multiplied their size contrary to the initial area after LVIs [22,23]. Such a scenario is especially dangerous if the damage in the element features as barely visible impact damage (BVID) or non-visible impact damage (NVID) [7].

The FMLs, materials more impact-resistant than FRPs [24], seem to be a good alternative. The limited damage area in the structure after impact and the presence of metal allowing it to absorb the energy by plastic strain give the FMLs potential to respond differently in CAI conditions.

The delamination itself does not change the cross-section dimensions of the FRP or FML; however, it significantly changes the mode of stress distribution and leads to local reduction of mode I fracture toughness. This is visible in buckling behaviour, as was proven by Hwang et al. [25], where three kinds of typical buckling response for FRPs due to compression were presented. The types were connected to the shape, size and location of the flaw in the structure. This means that the growth in initial damage can change load-carrying ability due to a shift in buckling force threshold.

The abovementioned phenomena are particularly important in slender elements. FMLs are mostly dedicated to thin-walled structures. Thus, the stability issues due to impact damage presence featuring as buckling should be expected and should be taken into account by analysing these materials [26]. Such material response promotes the propagation of delaminations caused by out-of-plane loads introduced while the half-waves are formed [27]. A combination of this fact with the risk of LVI damage leads to the extreme exposition of FRPs and FMLs to out-of-plane loads and finally to the increased risk of critical laminate failure due to extensive delamination growth.

The scarce (according to Rathnasabapathy’s thesis [28]) studies so far on FMLs subjected to CAI were able to present that the introduced impact damage is the cause of strength loss. Additionally, due to the used clamping conditions, the existence of buckling was reported in most of them. However, the damages were more likely to develop in the areas distant from the impact point, which made the explanation of impact influence on CAI behaviour unclear.

The first documented study on CAI of FML was made by Hoogsteden [29] on 1.35 mm thick 3/2 ARALL with the use of a Boeing grip. Due to stability issues, only the lowering of the buckling load along with the growth of impact energy was reported. The data presenting the impact damage propagation due to CAI were not included. Additionally, the reduction of maximal compressive force or strength was not given.

The study of Homan [30], (mentioned by Vlot and Gunnick [14]), does not present the specific test conditions or energy range used for inducing LVI damage in 5/4 GLARE. However, a 2% drop in compressive strength for plates with dents after impact and 10% for plates with cracks was observed. Kumar and Asokan [31] studied the reduction in compressive force transferred by the GLARE sample after impact; however, the amount of impact energy used was not given. The damage analysis was limited to the determination of recognized damage modes.

Jakubczak et al. [32] performed CAI tests on 2/1 GLARE (1.5 mm thick) by using an ASTM grip. The study revealed 17% strength reduction for 10 J impact energy. However, due to the free zone between the lower and upper part of the grip, the damage developed near the plate edge. Thus, the influence of impact damage on further propagation could not be evaluated.

Rathnasabapathy et al. [28] proved the increasing reduction of compressive strength along with the growth of impact energy. The greatest drop spotted in studied 2/1 GLARE (2.59 mm thick) secured by anti-buckling plates was equal to 22% for 19.5 J. Additionally, damage analysis based on the final specimen state was made. A further study [33] in similar conditions connected the use of prestress in an LVI event and CAI strength. For the highest impact energy and tensile prestress values, a 40% drop was reported. On the other hand, prestress by compression gave only 8% CAI strength reduction. In both pieces of research, the specimens tended to crush outside the anti-buckling plates.

Dhaliwal and Newaz [34] studied the 3/2 CARALL (2.3 mm) for LVI and CAI behaviour with an ASTM grip applied. A 42% of compressive strength drop was observed for 31 J of impact energy. However, only for the highest impact energy was the concentration of stresses in CAI seen around the impact area. Additionally, the dominant damage types for CAI were pointed out, but no information about damage growth related to the LVI damage area was given.

Bosbach et al. [35] tested CAI on 4/3 CARALL (5 mm thick); however, the reduction in compressive strength related to the intact state was not given, only the comparison to the carbon fibre-reinforced polymer (CFRP) counterpart (9% greater CAI strength of FML). Nevertheless, a detailed microscale damage analysis was carried out on the post-mortem sample, with numerous delaminations and inter-fibre failures identified. The effect of buckling was minor due to the large thickness of plates.

Jakubczak and Podolak [36] had already proven that the stress concentration within the impact area gives the best results for the FML while assessing post-impact residual properties and buckling behaviour. The same conception was checked earlier with success in the context of FRPs by Sanchez-Saez et al. [21]. Thus, such an approach was likewise implemented in this work.

None of the abovementioned research presents how the initial damage propagates inside the plate subjected to CAI or shows the after-failure state only. The answer to how LVI-induced damage progresses along the compressive force is still unknown. Additionally, the only correlation of CAI with the LVI event was given in terms of impact energy change. This work aims to present the dependence of CAI behaviour on impact energy and introduce damage size and other parameters describing the impact resistance of FML. The authors assume that such parametrical analysis supported by deformation analysis employing digital image correlation (DIC) will lead to the closure of detected gaps and clarification of hypotheses, where other factors were presented superficially. Additionally, the analysis of damage progression during the CAI may result in the detection of exact damage propagation time, description of complex damage forms and reasons for their presence. The work overall contains analyses enabling description of the ‘impact damage–buckling behaviour–damage growth’ relationship in FML, which has yet not been presented or discussed in the state of the art.

## 2. Materials and Methods

### 2.1. FML Manufacturing

The layup architecture of the studied FMLs is presented in Figure 1. The laminates were built of aluminium alloy sheets (0.3 mm thick 2024 T3, Alcoa, Pittsburgh, PA, USA) stacked alternately with glass unidirectional tapes for GLARE (AGL group) or with carbon unidirectional tapes for CARALL (ACL group) (S2-glass fibre or AS4-carbon fibre/135EP epoxy resin, 200 gsm, NTPT, Switzerland). The sheets were sand-blasted (Al_2_O_3_, 180 µm) and covered with sol–gel coating (AC-130-2, 3M, Saint Paul, MN, USA) to raise the metal–composite interface strength [37].

The curing of the 2.0 mm thick laminates stacked in a 3/2 configuration (Al/0/90/0/Al/0/90/0/Al) was performed in an autoclave (Scholz Maschinenbau, Coesfeld, Germany) [36]. To ensure the homogeneity of properties in the studied plates, a single process was performed on the 300 × 400 mm^2^ panels, which were later cut to the desired dimensions (100 × 150 mm ^2^).

### 2.2. LVI Tests and Damage Detection Techniques

The LVI tests were conducted with a set of different impact energies ($E_{0}$ of 4.5 J, 7.5 J and 16 J) on groups of five plates for each energy to enable comparison of the results for different conditions and the widest damage spectrum possible. Additionally, a set of five undamaged plates was used as a reference group in the CAI tests (Table 1). The experiment was performed on a SPNMS drop-weight impact tester. A 12.7 mm (1/2″) spherical impactor was used, to which a mass of 4.214 kg was attached. SPNMS software allows us to acquire force–time, force–displacement, and energy–time curves, from which the maximal force, damage initiation force (here interpreted as force value, for which the introduced damage decreased significantly the laminate bending stiffness), permanent displacement and value of absorbed energy were obtained.

After the LVI test, visual testing (VT) was done to describe the external changes in the FML, while internal damages were analysed based on the ultrasonic phased array through transmission (TTPA) technique [38]. By using pulser and receiver ultrasonic heads, 64 piezo-elements, 5 MHz frequency, defectoscope (OmniScan MX2, Olympus, Tokyo, Japan), and Tomoview 2.10R17 software, the C-scan projections of delaminations were obtained. Based on these, the damage area (DA) was calculated in image analysis software (ImageProPlus v. 6.0.0.260). The same procedures were repeated on the plates after CAI to describe the internal damage growth of FMLs.

### 2.3. CAI Protocol and DIC Implementation

The experimental stage dedicated to CAI was divided into two parts. The first was focused on the determination of CAI strength ($\sigma_{CAI}$), which equals the critical failure of the plate for impacted groups and the reference group. To present the damage propagation, three intermediate force levels, being the fraction of maximal force for the CAI test ($F_{CAI}$), were assigned for individual specimens—60%, 85% and 92.5% of force as the second part of the experiment. Such classification results from the need for presenting the plate behaviour in every stage of CAI response. After reaching the assumed thresholds, the plate was analysed with a focus on damages by using the procedure from LVI.

Detailed specification of the grip and justification of its usage is described in [36]. The plates were subjected to axial compression at the rate of 4 mm/s. The load was obtained via 8801 Testing Systems (Instron, High Wycombe, UK) and was able to reach 100 kN of force during the test (Figure 2).

The non-impacted side was observed during the test with the use of the DIC system ARAMIS 3D (Schneider S50 lenses, 800 mm adjustable bar, provided by GOM, Braunschweig, Germany). The pictures of the non-impacted side were recorded with the 20 Hz frequency by two 12 Mpix cameras (4096 × 3000 pix resolution) set to work in 280 × 205 × 205 mm^3^ volume (calibration deviation 0.067 pix, calibration deviation limit—0.100 pix). The GOM correlate (v. 2020 Hotfix 6) was applied to post-process the acquired data. From the recorded frames, the authors extracted the quantitative data—both for surface components (made of stochastic pattern painted on the specimen) and point components (markers on the measured objects—on the grip parts to compensate their motion about the specimen). The measured deformations were used to determine if and when buckling occurs, and what the magnitude of the phenomenon is.

## 3. Results and Discussion

### 3.1. Overview of LVI Results

The distribution of chosen parameters describing the FML response to LVI is presented in Figure 3. The increase in $E_{0}$ results in a significant increase in maximal force recorded on the LVI force–time curves ($F_{max}$) and permanent deflection of the plate after LVI ($d_{f}$) [11].

The force initiating significant damage of the plate during LVI ${(F}_{init})$ was determined for $E_{0}$ = 7.5 J and $E_{0}$ = 16 J. For 4.5 J of impact energy, $F_{max}$ was considered equal to $F_{init}$, as the plate displacement on force–displacement curves was still increasing after reaching this value. Figure 3 proves that AGL is more impact-resistant than ACL. AGL is characterized by greater $F_{init}$ (respectively, 2304 N and 2787 N in average), much higher $F_{max}$ (by 43% for $E_{0}$ = 16 J), and significantly decreased $d_{f}$ (by 18% for $E_{0}$ = 16 J) than ACL, despite the greater Young’s modulus of CFRP than glass fibre-reinforced polymer (GFRP) [39]. The ratio of $d_{f}$ to the maximal deflection of the plate (ACL—71% for $E_{0}$ = 16 J, AGL—61% for $E_{0}$ = 16 J) indicates the significant role of plastic deformation of metal while damage is developed.

With the increase in $E_{0}$, the energy absorbed by the plate due to LVI ($E_{abs}$) from the initial amount rises [10]. The ACL laminates are characterized by a higher $E_{abs}$ than the AGL (13.04 J—81.5% vs. 9.61 J—60%, respectively, for $E_{0}$ = 16 J). This fact is the consequence of a greater amount of damage introduced to the material [40]. The magnitude of DA for ACL is 1.5 ÷ 2 times greater than for the AGL (dependent on used $E_{0}$), which is strictly connected to differences in the fracture energy release rate [41] of each metal–composite interface, which affects mainly the extension of delaminations.

The differences presented above are related to the results of VT, which are compiled in Figure 4. VT resulted in detection of diversified damage forms related to FML structure and impact energies applied. In both of the groups, $E_{0}$ = 4.5 J creates BVID, featuring as the dent on the impacted side and bulge on the non-impacted side. $E_{0}$ growth caused only the presence of dents/bulges of greater size for the AGL group. However, a minor crack in the bottom metal layer, running along the 0° fibres, was detected for $E_{0}$ = 16 J.

For the ACL, a similar crack was found for $E_{0}$ = 4.5 J. The further increase in $E_{0}$ leads to the introduction of more complex damage forms for ACL. For $E_{0}$ = 7.5 J, a 0°-oriented crack of greater length than for $E_{0}$ = 4.5 J was spotted. Additionally, cracks running parallel to 90° fibres were detected—one with the origin in the middle of the in-length crack, the other being the lateral extension of the main crack. Such damage formation suggests that the impact energy used was close to that resulting in full perforation of the laminate. The action of $E_{0}$ = 16 J caused not only greater dents but also the fracture of the metal layer directly in contact with the impactor. Due to the high fragility of CFRP [42], the impactor moved freely through all plies of the FML and caused perforation and petalling of the bottom metal layer.

These observations made on the basis of parametrical analysis and visual inspection, which underline the influence of impact energy and laminate type, will be correlated with the description of the FML response under CAI.

### 3.2. Estimation of CAI Strength and Point-of-Stiffness Reduction of FMLs

The CAI experiment was conducted both on the reference group ($E_{0}$ = 0 J) of FML plates and the test groups (plates subjected to LVI). The relation of $\sigma_{CAI}$ (calculated on the basis of initial cross sections of the specimens) and $F_{CAI}$ to sample shortening during the CAI test ($d_{CAI})$ is presented in Figure 5.

Upon analysing the charts (Figure 5A,B), for the ACL group, a greater (~37%) $\sigma_{CAI}$ than for the AGL counterpart is seen for $E_{0}$ = 0 J. The difference is kept on a similar level (except for 16 J—22%) while comparing $\sigma_{CAI}$ for impacted groups. For both AGL and ACL (Figure 5C), strong and inverse correlations between $\sigma_{CAI}$ and $E_{0}$ (Pearson’s r coefficient for linear regression equal to −0.94 for ACL and −0.64 for AGL) were found [33,34]. A drop in $d_{CAI}$ following the growth of $E_{0}$ for both studied series was also observed (Figure 5A,B). Additionally, the effect of Young’s modulus can be seen on CAI curves—the $d_{CAI}$ for ACL is significantly lower than AGL ones.

The local differences between $\sigma_{CAI}$, i.e., differences between consequent $E_{0}$, were also considered. For the ACL group, the disparity of $\sigma_{CAI}$ and $d_{CAI}$ for intact plate and plate damaged by $E_{0}$ = 4.5 J is marginal. Nevertheless, the shape of the CAI curve is significantly changed, which suggests that the LVI damage affected the plate behaviour during CAI. Further, between the 4.5 J to 7.5 J pair, and 7.5 J to 16 J pair, major changes in $d_{CAI}$ and $\sigma_{CAI}$ are reported. As with the ACL group, minor changes in $\sigma_{CAI}$ and $d_{CAI}$ are visible between the intact plate and plate damaged by 4.5 J of impact energy in the AGL.

Detailed analysis of the $F_{CAI}{-d}_{CAI}$ curves for the ACL and AGL groups leads to a similar description of plate response. The curve trajectory can be divided into four phases of FML CAI behaviour (Figure 6). The proposed classification corresponds with the conventional response of FRP to CAI types of load [22,43,44]; however, some new and important factors can be attributed to FMLs.

Phase I is strictly connected with the establishment of contact between plate surfaces and surfaces of the CAI grip. Thus, no significant damage growth should be expected here. During Phase II, the initial stiffness of the FML is built and elastic response of the metal layers and composite layers assumed. Nevertheless, there is a potential to develop the delaminations’ size. Phase III leads to plate stiffness loss due to buckling. The additional action of lateral deflections gives us the basis to claim that the elastic–plastic response of metal layers can be visible and the beginning of fibre fracture due to CAI takes place. Additionally, the potential for damage increases significantly, especially through delaminations. In Phase IV, the accumulated energy is dissipated by plastic damage of metal, fracture of composite layers and sudden propagation of delaminations, which is visible through a rapid force decrease.

Through the analysis of the $F_{CAI}{-d}_{CAI}$ curves and designation of FML response phases, the critical points on the trajectories were identified from the point of view of CAI strength. The state of art terms FRPs “knee points” [23,43] to describe the stability loss of the plate. In fact, for the FMLs, the knee points could present the stiffness loss of the laminate caused by the extensive damage growth or buckling domination in the plate’s strain process. For the considered FML plates, two types of knee points were considered. The first (knee point type I) represents the end of the linear trend of the segment describing the initial stiffness of the plate [23]. The second (knee point type II) represents the intersection of two trend lines: the first for the initial stiffness and the second based on the segment of reduced stiffness [43]. An analysis of stiffness reduction of tested FMLs based on these approaches is presented in Figure 7.

Regardless of the studied material, the charts (Figure 7) confirmed that the increase in $E_{0}$ directly caused (high values of Pearson’s r coefficients) a decrease (negative r values) of the force needed to reach both type I and type II knee points. Thus, an increase in $E_{0}$ accelerates the stiffness loss presence of the impacted FML plate. The acceleration of the stiffness loss is greater for the ACL group by 24%. Independently of the assumed knee point, the force dedicated to ACL is significantly higher than for AGL (average 45% for type I and 72% for type II). The comparison attached above shows that both types of knee-point estimation are proper to describe the FMLs’ mechanical response under CAI.

To classify the studied materials, in terms of damage tolerance defined as the ability to withstand LVI and CAI jointly, the authors intended to accomplish it by more parameters than $\sigma_{CAI}$ only. The energy dissipation coefficient (EDC) was proposed, which is the ratio of overall absorbed energy (sum of $E_{abs}$ [J] from LVI and $W_{CAI}$ [J]—total work done on the initiation, propagation and relaxation of deformation and damage of FML during CAI) referred to the specimen volume (V) [cm^3^] (Equation (1)).
(1)$$EDC=\frac{E_{abs}+W_{CAI}}{V}\;[J/{\mathrm{cm}}^{3}]$$

The collation of EDC values for the plates is presented in Figure 8. The ACL is capable of accumulating more energy until failure under LVI and CAI together, due to its high stiffness of carbon fibre. However, when severe damage from LVI is induced, a lot of fibres no longer possess load-carrying ability. Summarizing the minor loss of $\sigma_{CAI}$ for the ACL with $E_{0}$ = 4.5 J and its relatively high EDC, one can state that in terms of residual properties, the BVID/NVID is not as dangerous as in the case of FRPs [45].

On the other hand, the EDC value for AGL with $E_{0}$ = 16 J is greater than for the lower $E_{0}$ and almost equal to the initial state of the FML plate. Potentially, this indicates the point of damage insignificance. Upon correlating EDC with low $\sigma_{CAI}$ loss, larger impact energies inducing more extensive and more complex damage forms are needed for the AGL group to lower the EDC value and CAI strength simultaneously than for the ACL laminates.

### 3.3. FML Deformation Analysis

The analysis of 3-dimensional deformations of FML during CAI allows one to estimate the effect of impact-induced damage. The data were compared later with the damage analysis results to correlate mechanical properties and material type. The deformations of cross sections running through the impact point projected on the non-impacted side were analysed (Figure 9). The analysis proved that every considered FML subjected to impact had buckled in the direction of impactor movement. Thus, the indentation created due to impact enforces the direction of plate buckling. There is a possibility for non-impacted plates to yield in the potential direction of impact likewise, but this is dependent on multiple factors e.g., structural defects of metal [46], structural defects of composite layers [47], microscale damages [26], geometrical imperfections [48] etc.

Another common feature is the distribution of the *Z*-axis deflections during CAI ($d_{buck}$)—a single halfwave is formed through the total loading process. The shape of the halfwaves was considered too—for the non-impacted FML plates and plates with $E_{0}$ = 4.5 J, an asymmetry was found. For higher $E_{0}$, the symmetric formation of the halfwave is visible. This may be a result of matrix and fibre fracture quantity combined with the relatively large DA, which enforces the presence of the greatest $d_{buck}$ close to the impact point. DA, determined by delaminations—especially those on the metal–composite interface (the weakest part of the FML structure [49])—causes the particular FML layers to move and pivot in more degrees of freedom while being in an enforced buckling state.

To evaluate the total deformation of the laminates in the impacted area due to LVI and CAI combined and to compare the overall material exertion, the $d_{p}$ parameter was created, which is the sum of $d_{f}$ and the maximal absolute value of $d_{buck}$ during the CAI, which was read from the defined cross section (Figure 9). The analysis of data depicted in Table 2 proved that despite slight (11% for ACL between the undamaged plate and $E_{0}$ = 16 J) or moderate (47% for AGL between the undamaged plate and $E_{0}$ = 16 J) decreases in $d_{buck}$ with increased $E_{0}$, $d_{p}$ still tended to grow alongside $E_{0}$. Additionally, the share of $d_{f}$ increased significantly with the use of higher-impact energies. This is mainly connected to the metal strain while FML was subjected to LVI. The hardening process due to plastic strain partially blocks the ability of metal in the impact area to deform further.

The damage forms present after LVI significantly affect the growth of $d_{p}$. For cases where BVID was reported, the increase in the parameter was up to 50% in comparison with the undamaged plates. For the ACL, at $E_{0}$ = 7.5 J and $E_{0}$ = 16 J, where accordingly severe damage and full perforation occurred, the rise in $d_{p}$ even reached 125%.

The analysis of deformation due to CAI of the overall free area of laminates is presented in Figure 10. The collation supports the conclusions drawn from the cross-section analysis. Regardless of whether the plate were damaged or not, the buckling initiates and propagates only around the central area. After reaching the critical failure state, the plate buckles significantly through its whole width, which was noted for both groups. This suggests a global buckling presence according to Hwang’s classification [25].

Upon tracing the displacement maps, the area of buckling before reaching the critical failure of the FML narrows with the increase in $E_{0}$—it is more focused in the impact area and suggests greatest stress concentration there. Such a state is kept in every analysed threshold, but still exceeding critical the point leads to buckling in the whole plate width.

The only difference seen between the ACL and AGL groups is the displacement distribution at critical failure. AGL tends to deform asymmetrically—such a scenario can be an effect of kink bands forming. The relatively low stiffness, shear strength and high elongation ability of glass fibre led to earlier and more intense presence of microdamage in fibres, following Budiansky’s model [50]. However, due to lower fracture toughness than for CFRP, the amount of energy released at failure does not lead to sudden crack growth in all directions, but to crack propagation towards one edge of the plate earlier than to the other one.

### 3.4. Analysis of Internal Damage Propagation

To determine the influence of impact energy on damage growth and final failure of FMLs due to CAI, ultrasonic C-scan mapping (TTPA method [38]) was performed. Figure 11 and Figure 12 present the projections of damage maps and changes in the delamination growth ratio (DGR) parameter, which is the percentage change of DA after CAI related to the DA after LVI.

In all of the cases, the main direction of delamination growth was along the width of the plate (90°). This was also spotted for the FRPs [23,43,51] and FMLs [34] in other studies. Another similarity with the FRPs is the difference between the initial DA and final DA—the value of DA changed from dozens to almost a hundred times, contrary to the initial DA caused by LVI.

A common tendency for both FML groups was spotted. The rise in DA during the CAI is insignificant even for the high percentage of $F_{CAI}$. This means that the whole damage propagation process held between the start of stiffness reduction and time of $F_{CAI}$ gain. The major difference between the ACL and AGL groups is the final DA: for the ACL, it is approx. 1.5 ÷ 3 times (dependent on $E_{0}$) greater than for the AGL. Additionally, the behaviour of both reinforcement types under LVI is different: the carbon fibres can break within the impact area even under lower-impact energies, so the areas more distant from the impact point have to carry the applied load and distribute stresses produced by CAI. Finally, this leads to greater stress concentration on a wider perimeter and the release of accumulated energy in a greater volume of the material.

The second reason for such differences between damage propagation in ACL and AGL is the fact that glass fibres possess a higher ability to deform than carbon ones. Initially, the assumed amount of fibre fracture of CFRP leads to the conclusion that the matrix fracture is intensive in a similar way and that this should not be expected for GFRP on such a scale. The matrix damage interrupts the proper transfer of stresses on adjacent fibres and has the potential to initiate delamination from the point of its presence.

This is directly related to the micromechanical behaviour discussed earlier. The tiny space created by delamination, allowing carbon fibres to move and pivot, enforces the shear stress creation in the matrix with the parallel existence of fibre kinking. This allows migration of the existing cracks further and enlarges the delamination. The properties of glass fibre allow the delay of crack migration in the matrix and layer-to-layer interfaces. Although the delaminations in AGL are limited, the above description supports that the presence of intense matrix cracks cannot be excluded.

Moreover, in the case of delaminations, mode I and mode II fractures should be considered. Abir et al. [52,53] and Bieniaś et al. [54] proved that the properties connected to interlaminar crack initiation and propagation are crucial for further delamination development. As the deformation analysis proved that the global buckling mode occurred during the CAI, it can be assumed that mainly the mechanisms connected to the mode II fracture were present.

Due to out-of-plane loads resulting from geometry deviation after stability loss, the shear stresses were accumulated on the interfaces between each layer pair of the laminates. The minor increase in DA after each threshold of force underlines that the interlaminar properties of studied laminates were sufficient to withstand the considered conditions and accumulate energy until the stage of critical failure. After exceeding the critical load, the buckling mode became impossible to define clearly. The mixed buckling mode is related to the release of the energy accumulated in the interfaces by means of mode II fracture. After the propagation of delaminations as the effect of energy release, the influence of mode I fracture can be observed as each layer individually can buckle in different directions, causing the interlaminar crack opening, but still, the main direction of lateral deformations of FMLs is possible to determine.

### 3.5. Macroscopic Observation

The final damage to FMLs resulting from compression after impact (full CAI loading process, as well as partial CAI loads) was analysed based on macroscopic observation. Images of ACL and AGL laminates are presented in Figure 13. The analysis of the damaged laminates showed that regardless of fibre type, the final form of failure is similar. The dominant damage type is plastic deformation of metal layers—mainly focused on the unconstrained parts of the plate. If considering the damage maps from TTPA analysis (Figure 11 and Figure 12), the delaminations should be categorised as the dominant damage type as well (fitting approximately into the potential damage zone). For the plates used to check the intermediate force thresholds, no characteristic change in the external look was detected.

On the global scale, the effect of impact existence is minor—it only enforces the direction of buckling and stress concentration around the damaged area. The only disparity was detected for the AGL plates, where damage after CAI was distributed asymmetrically in some of the cases. Additionally, the growth in external signs of post-impact damages (Figure 13B,D) was minor or did not happen at all (even upon analysing metal cracks on the bottom layers). The integrity of the laminate was relatively retained—i.e., the compressed plate was not physically divided into multiple parts due to critical failure. As the composite layers individually were expected to be fractured into pieces [51], the metal layers kept the whole part together. However, differences were spotted in the matter of local changes in the studied plates, which are presented in Figure 14.

A common feature of the analysed areas is the tendency to form fibre cracks in the direct vicinity of permanent metal deformation or fracture. The places of metal fracture, fibre cracks or matrix cracks are linked by delaminations, both on the metal–composite interface and composite–composite interface. Additionally, matrix cracks tend to form in the 90° layers of the composite between adjacent fibres, where the crack can develop in a translaminar way, joining the eventual delaminations on 0°–90° interfaces or metal–composite interfaces, while the integrity of 0° fibres is breached.

Another factor is the amount of impact energy used to initially damage the plates. Upon analysing the close-ups from Figure 14, one can state that the growth in impact energy makes the propagation of delaminations more likely than the creation of new fibre or matrix cracks. The growth in impact energy decreases the curvature radius of deformed metal too. This is caused by the stress concentration around the impact area, which is presented in DIC displacement maps (Figure 10), where areas near the lateral surface are subjected to lower strains.

The difference between the applied reinforcement types is significant. First, the presence of rigid and fragile carbon fibre induces the occurrence of extensive metal layer deformations, which lead to significant fractures of the aluminium if considering the undamaged plate and plate with BVID. Such behaviour was not recognized for the AGL in any studied specimen. Another disparity is the radius of metal layer curvatures: for the more deformable glass fibre, the transitions between bending points are smoother.

Fragility is also visible in the quantity of fibre and matrix cracks visible in relation to AGL. Additionally, due to poor values of fracture energy release rate for mode II [54], more metal–composite delaminations should be expected in the ACL group. The same was found for the composite–composite interfaces—the failure of this FML part is greater for ACL too.

### 3.6. The Solution of FMLs’ CAI Strength Improvement Based on Damage Growth Analysis

The conducted research confirms that the impact damage is the main factor decreasing the compressive strength of thin-walled composite structures, including fibre metal laminates. Along with other studies [28,33,34], this research also confirms the strong correlation between the impact energy and CAI strength of the FML.

Upon comparing the results from [34], where weaker aluminium alloy (5052 H32) was used, with the results described in this paper, the authors state that one of the ways to improve the residual strength of impacted FMLs is the application of aluminium alloys with higher yield strength and elongation values (e.g., 6061 T6, 2024 T3, 7075 T6). The alloys with other elements being the alloy matrix should be considered too (e.g., titanium alloys, stainless steel). First, such a step would significantly increase the energy absorption from the LVI event. Secondly, this would raise the strain needed for failure, which is especially important for keeping the integrity of the part after reaching critical stresses.

Our results concur with conclusions made by other authors [33,34,35], indicating the need to increase the interlaminar properties of the laminate to limit the delamination size. This research supports that further by analysis of damage development in studied thresholds. The insignificant increase in the damaged area even for the high forces means that the energy is mainly accumulated in the material and released through the sudden delamination growth and fibre and matrix crack occurrence in the moment of critical failure. However, the use of matrices with higher static properties is also encouraged from the point of view of initial delamination size, damages induced by impact and mechanism of kink bands [55,56] forming under compression.

Nevertheless, one can state that the combination of elastic–plastic metal layers, fibre-matrix composite layers and their mechanochemical [9] connection results in a significant advantage for FMLs in terms of CAI strength compared to conventional composites (e.g., [51]). The main reason for that is the high static strength and limitation of rapid buckling of the delamination after impact.

## 4. Conclusions

This work concludes the experimental research on aluminium-based FMLs subjected to LVI and CAI. Apart from changes in $\sigma_{CAI}$ and damage development in relation to $E_{0}$ and used material type, the intermediate-force levels before critical failure were analysed to assess the dynamics of damage propagation due to CAI. The correlation of the CAI behaviour of FML with other LVI parameters was done likewise. Due to the LVI event, $\sigma_{CAI}$ of the FML drops, independently of the reinforcement type used. The increase in $E_{0}$ strongly affects not only $\sigma_{CAI}$ but also lowers knee-point force (10% ÷ 25%) and leads to greater stress concentration around the impact area during the CAI test.

GLARE is characterized by a higher ability to preserve its properties than CARALL (6% vs. 17% for $E_{0}$ = 16 J). However, CARALL is characterized by 30% greater energy dissipation coefficient for LVI and CAI together.

The final shape and magnitude of the halfwave formed due to buckling are dependent not only on $E_{0}$ but also the post-impact damage state. The magnitude of total deformation in the impact area resulting from LVI and CAI combined can be used as a comparison parameter describing overall material strain.

As the main damage modes, delaminations and plastic metal strain were found. The presence of metal fracture is more likely visible for CARALL than for GLARE. Additionally, the increase in $E_{0}$ promotes the growth of delaminations at the cost of insignificantly limited fracture of fibre and matrix.

The analysis of the damage area before and after CAI for critical failure and assumed force thresholds proved that significant damage propagation (up to 100 times) occurs during gradual stiffness loss. This means that the material mainly accumulates the energy and releases it after significant stability loss.

According to the mechanical and damage analysis, the existence of BVID/NVID in the FML seems to be insignificant from the point of view of property preservation. Contrary to the FRPs, the presence of metal layers changes the plate behaviour significantly—both in a mechanical way and in terms of damage initiation and propagation.

Based on this research, the following ways to improve damage tolerance of aluminium-based FMLs are proposed: (1) use of metal with the highest possible yield strength to increase energy absorption and overall CAI strength, (2) use of matrices with high mechanical properties to limit the kink bands forming during compression to inhibit the moment of stability loss, and (3) application of matrix types and metal surface preparation techniques promoting increased fracture toughness of the interfaces, especially for mode II fracture.

## Figures and Tables

**Figure 1 materials-16-03224-f001:**
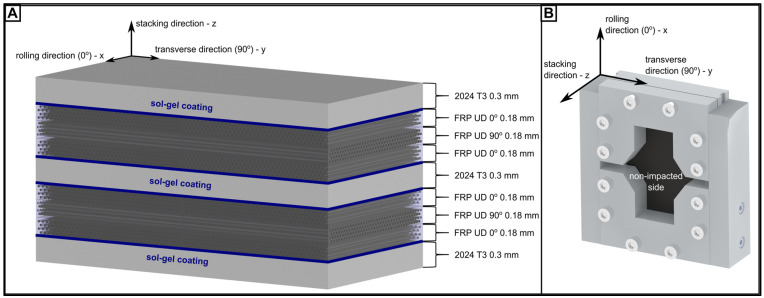
The structure of the studied FMLs for CARALL (**A**) and scheme of rig used for CAI test (**B**).

**Figure 2 materials-16-03224-f002:**
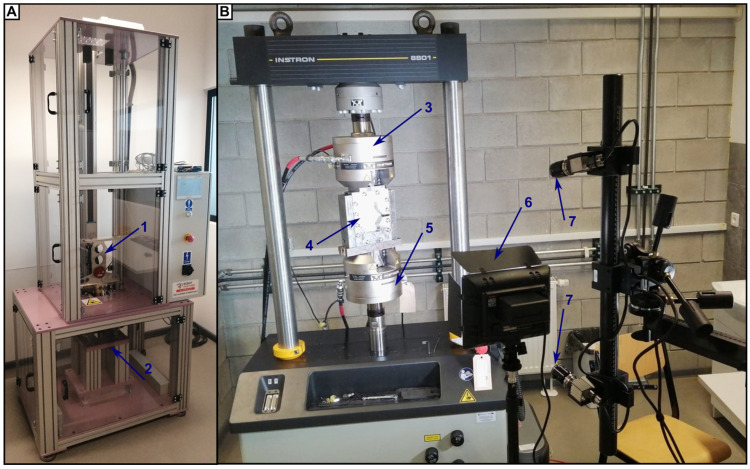
Presentation of used test stands for LVI tests (**A**) and CAI tests (**B**). The following features are marked: (1) impactor with dropping mass attached, (2) platform, (3) force registration cell, (4) CAI grip with mounted plate, (5) force generation cell, (6) light source, (7) DIC cameras.

**Figure 3 materials-16-03224-f003:**
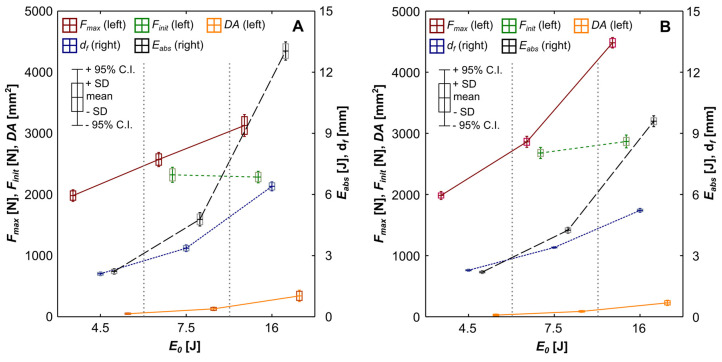
Physical parameters of low-velocity-impact phenomenon for ACL (**A**) and AGL (**B**). The values of maximal impact force ($F_{max}$), force of damage initiation ${(F}_{init})$, damaged area (DA), permanent deflection of plate ($d_{f}$) and absorbed energy ($E_{abs}$) are shown as box plots to simplify statistical comparison.

**Figure 4 materials-16-03224-f004:**
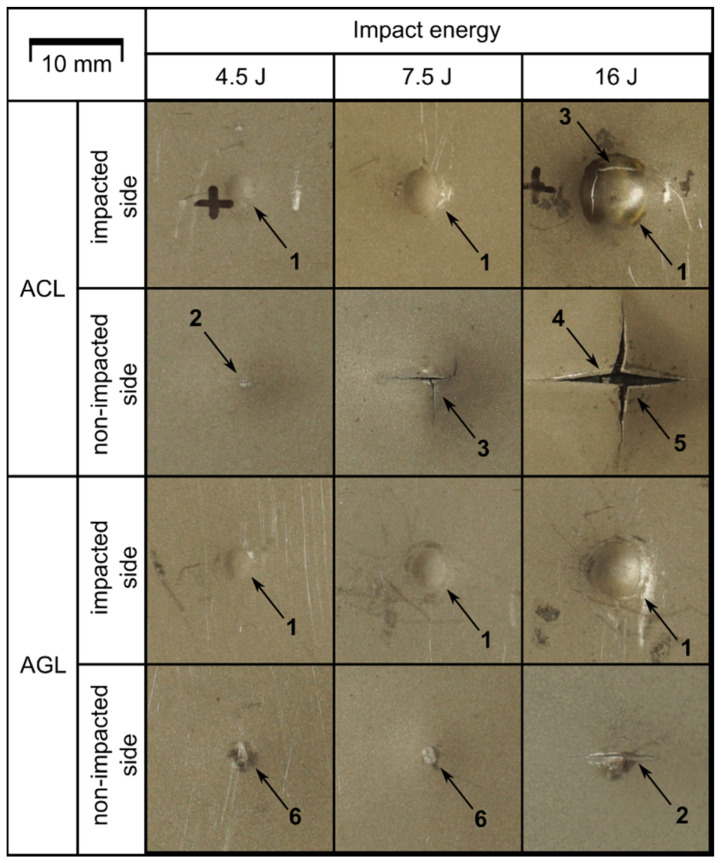
Comparison of FML plates damaged by LVI: close-ups of impact area on impacted and non-impacted sides. For the analysed plates, the following forms of damage were recognized: impact dent (1), minor cracks of metal layer (2), extensive cracks of metal layer (3), fibre cracks (4), petalling (5), plastic deformation of bottom metal layer (6).

**Figure 5 materials-16-03224-f005:**
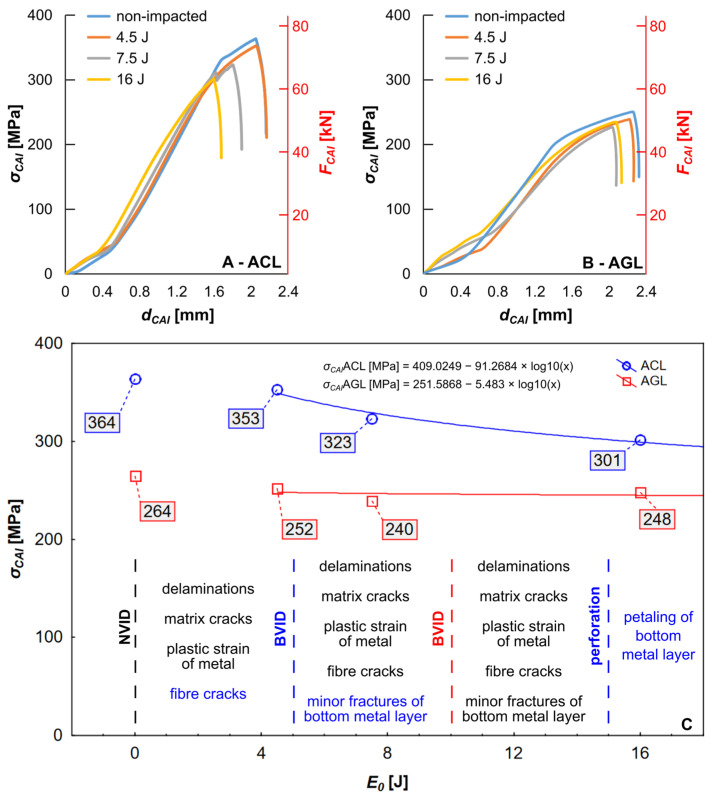
Force ($F_{CAI}$)/strength ($\sigma_{CAI})$–displacement ($d_{CAI}$) characteristics of tested FMLs under compression after impact ((**A**)—ACL, (**B**)—AGL). The strength was correlated with the impact energy $\left(E_{0}\right)$ and damage forms recognized after low-velocity impact to present the influence of impact condition on residual strength of FML (**C**).

**Figure 6 materials-16-03224-f006:**
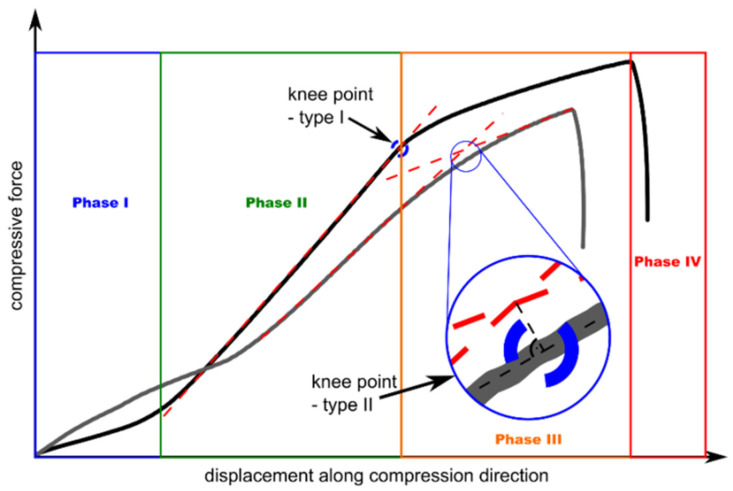
Phases of the FMLs’ mechanical response while subjected to CAI and methods of knee-point estimations. Phase I—system stabilization, Phase II—linear force growth, Phase III—gradual stiffness degradation, Phase IV—critical failure.

**Figure 7 materials-16-03224-f007:**
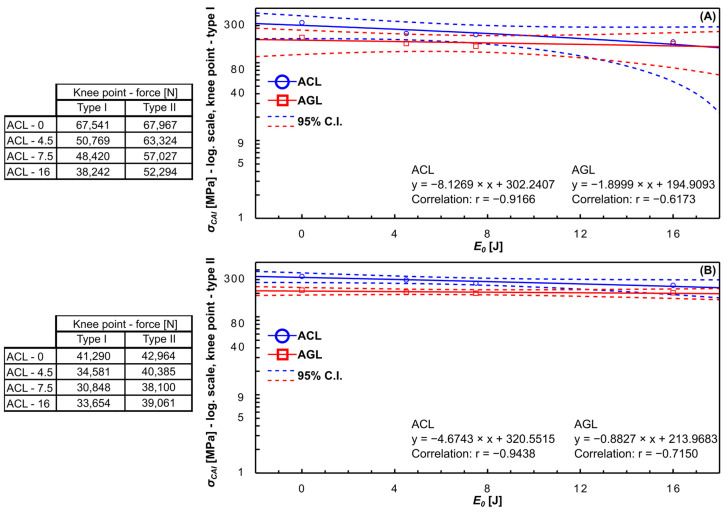
The relation of impact energy and knee point force/stress values detected on CAI test characteristics for type I (**A**) and type II estimation methods (**B**).

**Figure 8 materials-16-03224-f008:**
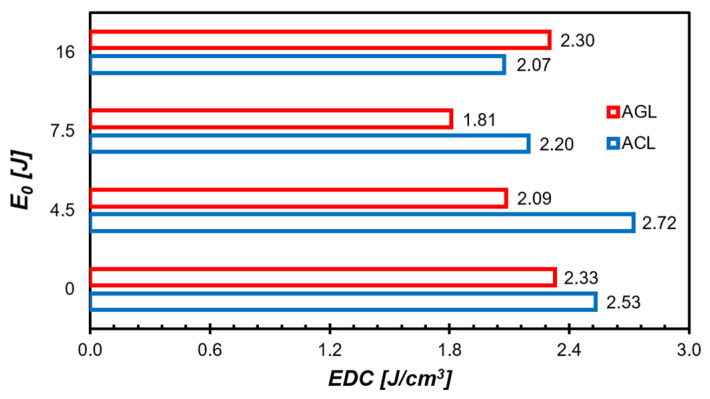
Comparison of EDC values for studied FML plates. The ACL laminates tend to dissipate more energy per volume unit than AGL laminates.

**Figure 9 materials-16-03224-f009:**
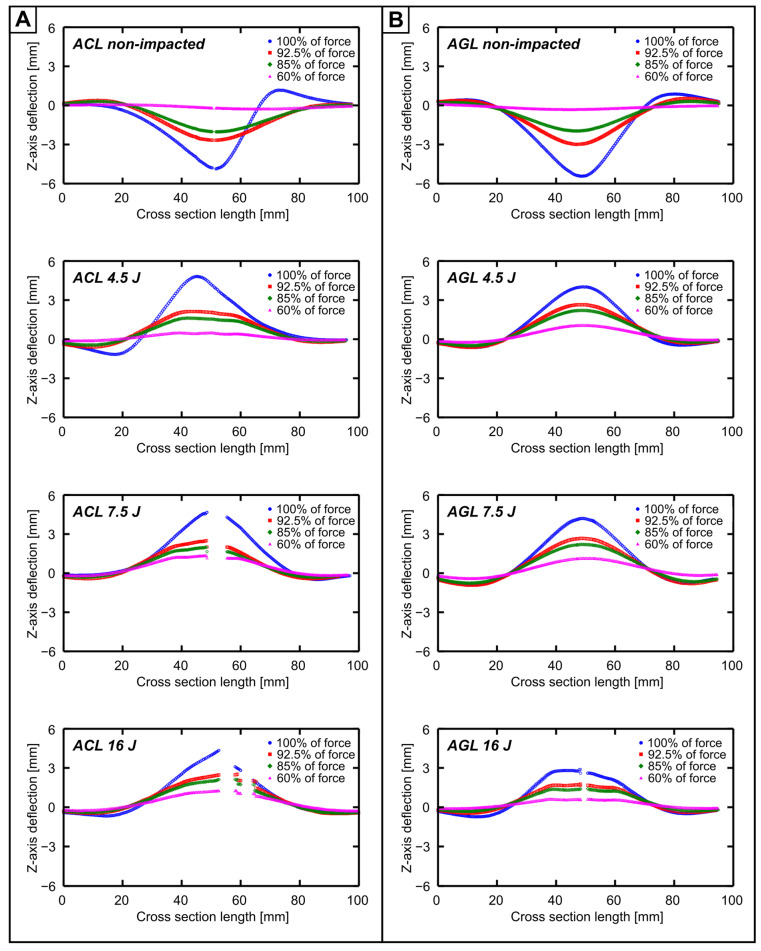
Comparison of displacement distribution on the plate cross section running through the impact point. Deformations for ACL (**A**) and AGL (**B**) were correlated with impact energy and used force threshold.

**Figure 10 materials-16-03224-f010:**
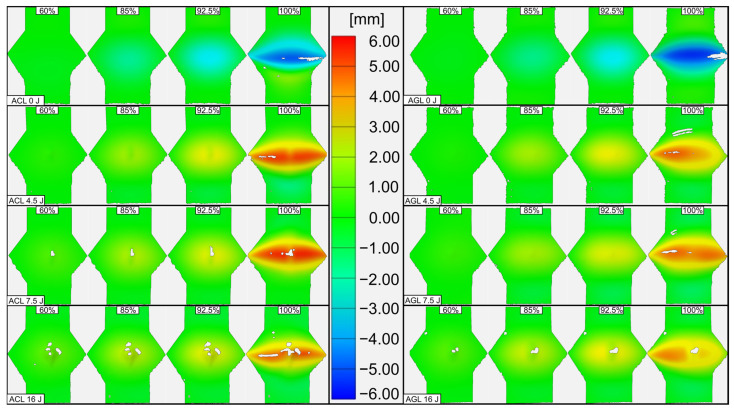
*Z*-axis deformation maps of compressed plates registered by DIC system. Displacement was the most visible after reaching the buckling state during the CAI test.

**Figure 11 materials-16-03224-f011:**
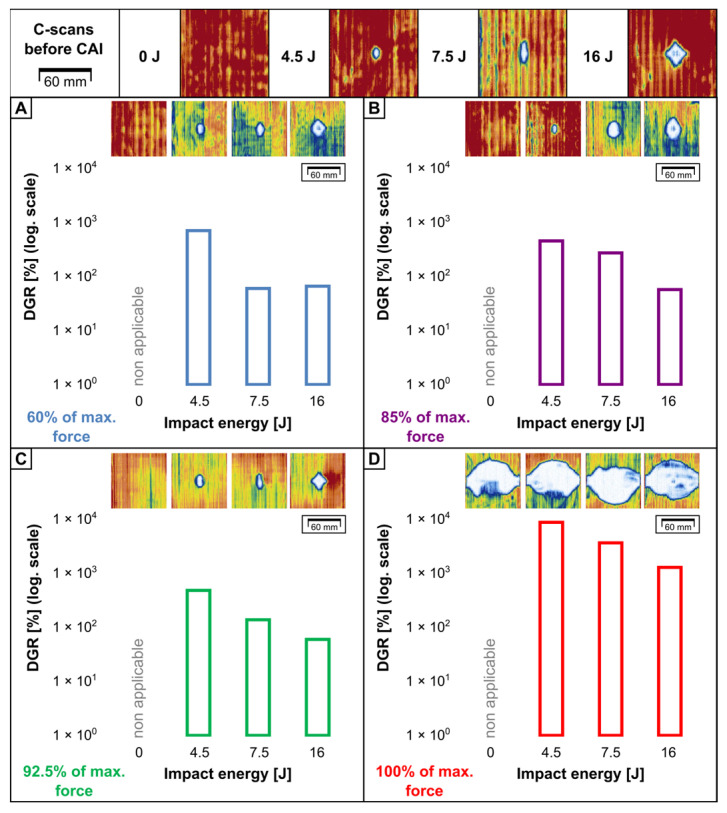
Ultrasonic C-scans of damaged areas in ACL laminates before CAI and after CAI for following force thresholds: 60% (**A**), 85% (**B**), 92.5% (**C**) and maximal force (**D**). Also the relation of delamination growth ratio to impact energy was presented.

**Figure 12 materials-16-03224-f012:**
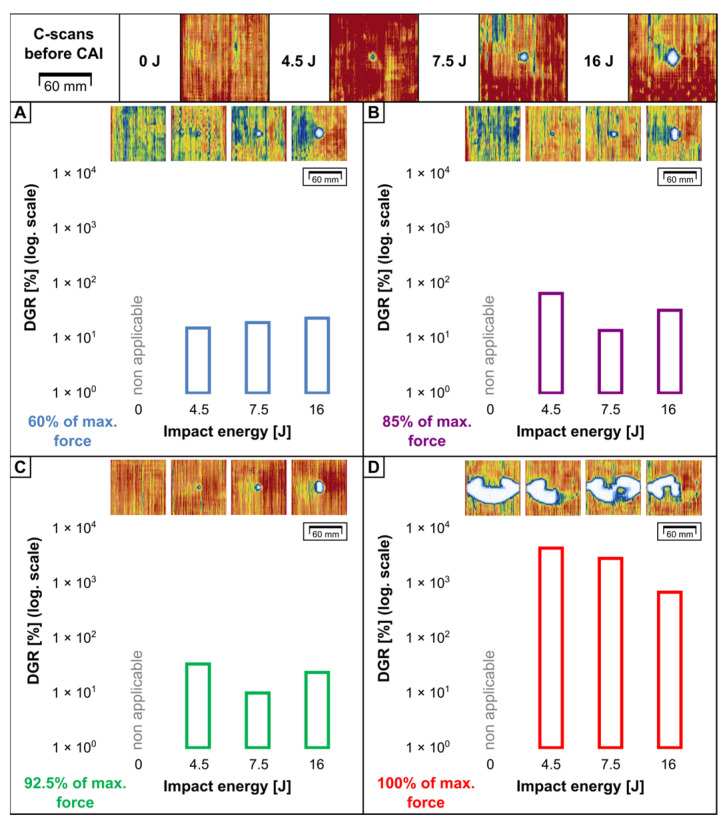
Ultrasonic C-scans of damaged areas in AGL laminates before CAI and after CAI for following force thresholds: 60% (**A**), 85% (**B**), 92.5% (**C**) and maximal force (**D**). The relation of delamination growth ratio to impact energy is also presented.

**Figure 13 materials-16-03224-f013:**
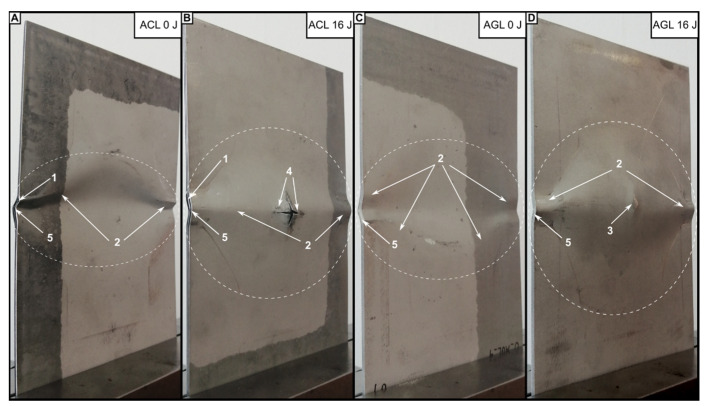
Macroscopic views of ACL (**A**,**B**) and AGL (**C**,**D**) laminates after critical failure in CAI test. The following damage forms were recognized: delaminations (1), plastic metal strain (2), metal cracks (3), post-impact petalling (4) composite layer failure (5). The perimeter of potential damage zone is marked with dashed lines.

**Figure 14 materials-16-03224-f014:**
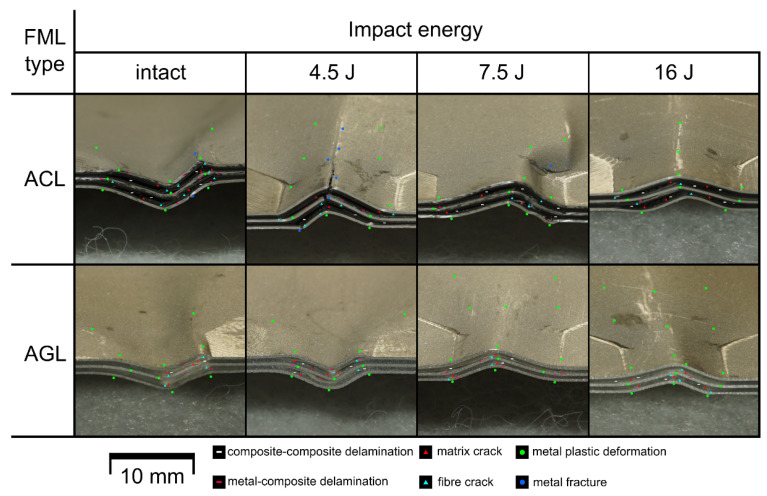
Detailed analysis of damage of FML after full CAI loading process.

**Table 1 materials-16-03224-t001:** Designation of FML plates prepared for LVI and CAI tests.

Impact Energy	0 J	4.5 J	7.5 J	16 J
ACL (aluminium/carbon fibre laminate)	ACL-0-1	ACL-4.5-1	ACL-7.5-1	ACL-16-1
	ACL-0-2	ACL-4.5-2	ACL-7.5-2	ACL-16-2
	ACL-0-3	ACL-4.5-3	ACL-7.5-3	ACL-16-3
	ACL-0-4	ACL-4.5-4	ACL-7.5-4	ACL-16-4
	ACL-0-5	ACL-4.5-5	ACL-7.5-5	ACL-16-5
AGL (aluminium/glass fibre laminate)	AGL-0-1	AGL-4.5-1	AGL-7.5-1	AGL-16-1
	AGL-0-2	AGL-4.5-2	AGL-7.5-2	AGL-16-2
	AGL-0-3	AGL-4.5-3	AGL-7.5-3	AGL-16-3
	AGL-0-4	AGL-4.5-4	AGL-7.5-4	AGL-16-4
	AGL-0-5	AGL-4.5-5	AGL-7.5-5	AGL-16-5

**Table 2 materials-16-03224-t002:** Deformation parameters of ACL and AGL laminates caused by combined impact and CAI.

Material	$E_{0}$	*d_f_* [mm]	$d_{buck}$ [mm]	$d_{p}$ [mm]	Short VT Description after LVI
ACL	0	0	4.87	4.87	undamaged
	4.5	2.08	4.82	6.90	visible dent, minor fracture of bottom metal layer (BVID)
	7.5	3.20	4.67	9.87	severe fracture of bottom metal layer, onset of petalling
	16	6.60	4.34	10.94	petalling of bottom metal layer, full perforation
AGL	0	0	5.43	5.43	undamaged
	4.5	2.29	4.02	6.31	visible dent (BVID)
	7.5	3.39	4.21	7.60	visible dent (BVID)
	16	5.22	2.89	8.11	visible dent, minor fracture of bottom metal layer (BVID)

## Data Availability

The data presented in this study are available on request from the corresponding author.

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
