# Peer review of "The Correlation of LVI Parameters and CAI Behaviour in Aluminium-Based FML"

_materials, 2023, doi:10.3390/ma16083224_

Round 1

Reviewer 1 Report

In the current study, experimental analysis of mechanical behaviour for aluminium-based fibre metal laminates under compression after impact was conducted. The topic is original and fills the gap of  greater energy dissipation ability of LMPs. A better explanation of  energy dissipation ability of LMPs has been supported by experimental data.

Although the work is interesting but 10 self citations have been observed in the article. Authors are suggested to do maximum of 4, 5 self citations and to add relevant references in place of self cited references.

Reviewer 2 Report

The paper is of interest but some changes are needed before publication

-Line 146: It is not necessary to put the units between [ ], i.e.: mm

-Subsection 2.2. Please provide a table explaining all configurations and nomenclatures to clarify the methodology.

-An image of the general setup of the test should be added

-If the applied energies were 4.5, 7.5 and 16 J, why do the parameters in Figure 2 not appear aligned to these energies? Please clarify. It is very strange.

-Fig 2: If for 4.5 J there is no damaged area and the Finit value is not reached, how can there be a permanent deformation of the plate?

-Please clarify what authors consider as "significant damage" to establish Finit. This should be explain in Materials and methods section.

Please unifiy the format of Figures. Figure 7 format (axis, etc.) is totally different from Figure 6 or 4

The results section is somewhat lengthy. There are parts that repeat themselves or say basically the same thing. The whole section should be revised.

Reviewer 3 Report

In this paper, the authors experimentally and numerically the mechanical behaviour for aluminium-based fibre metal laminates under compression after impact. Damage initiation and propagation were evaluated for critical state and force thresholds. It is shown that the relatively low-energy impact has marginal effect on fibre metal laminates compressive strength. Metal plastic strain and delaminations are dominant failure modes for compression after impact. The manuscript is very interesting and up-to-date, therefore worth to publish.

However, the authors should consider the followings:

1. Please refine the Conclusion Section and present the important conclusion.

2. In Introduction, two publications on the FML structures under low-velocity impact should be added, bringing more information to readers.

(1) Low-velocity impact of sandwich beams with fibre-metal laminate face-sheets. Composites Science and Technology, 168 (2018) 152-159.

(2) Low-velocity impact on square sandwich plates with fibre-metal laminate face-sheets: Analytical and numerical research. Composite Structures, 259 (2021) 113461.

3. Please indicate the "sol-gel coating" clearly in Fig. 1(a).

4. How to verify the correctness of the results in Fig. 2? For example, the comparison between experimental and numerical results. 

5. In Fig. 8, why are curves discontinuous?

Round 2

Reviewer 3 Report

The manuscript can be accepted.